# Dysfunctional Schemas from Preadolescence as One Major Avenue by Which Meaning Has Impact on Mental Health

**DOI:** 10.3390/ijerph20136225

**Published:** 2023-06-25

**Authors:** Nathalie André, Roy F. Baumeister

**Affiliations:** 1Research Centre on Cognition and Learning (CeRCA), UMR CNRS 7295, University of Poitiers, 86000 Poitiers, France; 2School of Psychology, University of Queensland, St. Lucia, QLD 4072, Australia; r.baumeister@uq.edu.au

**Keywords:** schema, meaning, therapy, maladaptive schema, preadolescence, mental illness

## Abstract

A main way by which meaning influences mental health is by the formation of interpersonal schemas that specify what to expect from others and how to treat them. Particularly during preadolescence (a developmental phase focused on interpersonal skills), young people living in a stressful or hurtful environment can form atypical schemas that can help them survive but that produce serious problems when later applied to newly forming adult relationships. We provide three case studies illustrating this process. A boy learned to cope by withdrawing from social interaction and excelling in schoolwork. A girl learned to cope by denying her own needs and sacrificing herself for the welfare of others. Another girl coped by pervasive distrust of others and by becoming assertively independent. These children learned well enough to adapt to these dysfunctional relationships so as to suffer as little as possible, and they even developed some personal skills and resources. However, the rigid schemas had a destructive impact on their adult relationships. Proposals for interventions to change meaning and behaviors are discussed.

## 1. Introduction: Schemas as Meaning

The issue of meaning in life was once considered too abstract, subjective, and unscientific for scientists to study (most obviously in the days of rat labs). In recent decades, researchers have developed tools and theories to study it. The vast majority of such research has involved measuring (or occasionally manipulating) a person’s global feeling of meaningfulness in life. Essentially, it compares people who report their lives as highly meaningful against those who do not, typically with one number on a scale to represent the person’s perceived meaningfulness for his or her entire life. Meaning is treated as a quantity, so roughly all meanings are the same in that sense. All lives have the same kind of meaning when it is measured as a single number, or even a few. While we fully respect and appreciate the contributions of that approach, this paper takes a different and more nuanced approach. We examine how specific meanings form as ways of coping with difficult relationships or other life circumstances and how the resulting schemas can hamper adult functioning.

In this article, we focus on the schema as a potent form of meaning, consistent with the term’s usage in cognitive, developmental, clinical, and sociological studies [1,2,3,4]. People organize, comprehend, store, and relate important experiences in their lives through schematic representations. Schemas enable people to organize and represent the events they experience as meaningful wholes, enabling comprehension and guiding behavior. We focus particularly on schemas that can be adaptive ways of coping with an atypical and stressful interpersonal environment, which, however, proves maladaptive when subsequently applied to adult relationships. A clinician can help guide the reinterpretation and restructuring of the schema, possibly even transforming or replacing it, so as to foster behavior change and improve mental health.

## 2. Definitions of Schema and Meaning

The term schema is used in different fields of psychology (social, cognitive, developmental, and clinical). Despite possible subtle differences, all of them use the term to refer to a meaningful and organized assembly of concepts that helps to understand. We follow Beck [5] in using the term to designate basic knowledge structures, such as beliefs, that constitute a person’s understanding of self and the world. Relationship schemas are a person’s built-up interpretive understanding of how relationships work, including such things as how to expect to be treated by others, how oneself should act vis-à-vis others, and what causes relationships to go well or badly. A relationship schema may be specific to a particular relationship or may apply to a broad category of relationships (e.g., how to deal with co-workers, how to obtain and sustain romantic love). Schemas are subjective theories of everyday life, which explain how the world works. Indeed, one could doubt whether life could have meaning without schemas.

Meaning is famously difficult to define. While editing a special journal issue devoted to the psychology of meaning, Baumeister and Landau [6] observed two broad thematic properties. One is a nonphysical connection. Meaning links together items that may have no physical connection, as in the example of a flag representing a far-off country. The use of meaning enables the mind to connect different items, including nonphysical ones (e.g., ideas), in multiple ways. Second, meaning is a potential organization. While very simple meaningful thoughts such as associations merely connect two stimuli, more complex meanings such as systems, theories, and narratives (stories) can impose organizational structure on large amounts of information. 

Simple animal minds use meaning to connect and organize, and indeed, much of the animal learning literature focuses on associations. Human minds evolved to use meaning collectively, such as with language and shared understandings, which greatly increases the power of meaning [7]. Much information in the human mind is not there as the result of direct experience but comes via communications from others. 

## 3. Acquisition of Dysfunctional Schemas

Schemas form naturally in the human mind as it seeks to understand events by the use of meaning. Categories enable sorting, distinctions define something in relation to what it is not (e.g., wet is not dry), and associations link stimuli. The chaos of experience starts to make sense as the mind uses meaning to impose organization on it. Narratives, for example, are defined by collections of events spread across time in an organized fashion (sometimes with interloping periods irrelevant to the narrative, such as when an ongoing work project is temporarily suspended during a holiday), and also with a causal structure such as that one event leads to the next. 

Importantly, schemas enable the person to generalize about new situations. They are more than just an interpretation of a particular event. Rather, they are subjective theories of everyday life, which explain how the world works. For example, a man’s understanding of how to have a romantic relationship with a woman may be shaped by earlier relationship experiences and refined over the course of multiple relationships with different women. To each new relationship, he brings the schema developed from the previous ones. Ideally, he would revise the schema with each new partner; however, once a schema has formed, it tends to resist change [8]. As a result, his new relationship may be shaped, sometimes unfairly and inappropriately, even destructively, by the schema left from previous ones. Colloquially, this is sometimes called “emotional baggage,” a metaphor expressing the view that the person enters a new relationship carrying problems, concerns, sensitivities, and coping styles from previous relationships. 

The organizing function of schemas includes specifying what aspects of events are central and which are not [9,10]. Once formed and then activated in a new situation, schemas can set off automatic responses, including emotions, thoughts, and actions. Thus, a schema imposes a ready-made interpretation on novel events and prescribes how to respond.

Our particular focus is on relationship schemas formed in late childhood while coping with difficult or stressful interpersonal environments. Indeed, after choosing our case histories, we noticed that all three had their apparent origins during this phase of late childhood (rough ages 8 to 12). This phase was disparaged by Freud as “latency,” that is, a period in which psychosexual development largely went underground and ceased to make progress. In contrast, subsequent theorists who focused more on interpersonal processes emphasized these years as ones in which the growing child’s styles of interacting with others and carrying on relationships would be formed [11].

Relationships are important, and one of the biggest sources of meaning in life, but not all relationships work well, and even good relationships encounter periodic conflicts and problems. People develop meaningful interpretations (particular schemas) of themselves and others in order to cope with these difficulties. They may then apply these schemas to other, subsequent relationships. 

In older children and adolescents, the development of schemas is based on the satisfaction of emotional needs [12]. Among these needs, Young et al. [12] identified the following: secure attachments to others (including safety, stability, nurturance, and acceptance), autonomy, competence, and sense of identity, freedom to express valid needs and emotions, spontaneity and play, realistic limits, and self-control. Preadolescence characterizes the last years of childhood, between 8 and 12 years old, and refers to the period where early dysfunctional schema can develop. According to Young’s early dysfunctional schemas theory, preadolescents confronted with interpersonal contexts in which emotional needs are not met develop strong beliefs about themselves and the world. Schemas become more abstract, complex, organized, dense, and rigid as experiences accumulate. Hence, they become increasingly difficult to change. 

Although Young et al. [12] envision that certain schemas can be adapted to the environment and foster relevant coping styles, they proposed that many schemas become maladaptive and inaccurate as the child grows. We emphasize that the initial schema formation may be highly adaptive for helping the child cope with a dysfunctional family context or social environment. They mainly become maladaptive when the social environment changes and when the person seeks to build new relationships based on the model adapted to the previous, troubled ones.

It is tempting to label schemas as good or bad, but we note that the line is often blurred. Coping with difficult situations may cause the person to develop resources or abilities that could prove useful and helpful in other settings.

Schema-based therapy belongs among cognitive behavioral therapies. Under the influence of life events, dysfunctional schemas are activated and cause strong negative emotions such as shame, sadness, anger, and fear. The therapy proceeds by addressing the dysfunctional schema, helping the patients to understand their earlier relationships (from which the dysfunctional schema emerged). Often, patients can understand what happened when they were young, but they do not typically make the connection to their current problems in adulthood. Mindfulness meditation, reparenting, visual imagery, cognitive restructuring, and observation of emotions are generally the most used modes of intervention. However, the patient is sometimes reluctant to change because the schema is very embedded and automatic. Observing behaviors that validate and activate schemas can be an effective intervention to produce change.

## 4. Three Case Studies

### 4.1. Method

We present three case studies from the first author’s clinical practice in the past five years. The patients entered therapy for personal help and not for research purposes. They were chosen to exemplify the pattern of maladaptive schemas formed in response to one relationship but then damaging to subsequent relationships.

All three patients signed consent forms indicating their agreement that information about their cases could be used in publication anonymously. The names have been changed, and the cases are presented in a sufficiently impersonal fashion so that no one could ascertain the identities of the patients. 

### 4.2. Case 1: Paul Hides from Intimacy (But Excels in School and Work)

A 33-year-old man presents for clinical treatment for social anxiety. He tells the therapist that he has been anxious in his entire life. He grew up in a family with much hostility and conflict. His parents divorced when Paul was six. The father was jealous and abusive, and he never accepted the divorce, continuing to make romantic and sexual advances toward his ex-wife. 

Following the divorce, Paul lived with his mother and siblings. His mother soon took a new lover, a man whom Paul hated. Paul began to isolate himself from the family, spending much time alone in his room. He justified this by doing homework and reading. One result was that he became the only one among his siblings to succeed in school. However, the situation was unhappy. Paul, therefore, chose to switch and live with his father. The father was not welcoming, however. Paul recalled his father as generally cynical and sarcastic, both in his general outlook on life and in his treatment of his son. His father was also prone to rages. Paul developed a schema for dealing with these rages by being completely passive and by avoiding conflict as much as possible. Staying alone in his room continued to be an appealing way of coping. He continued to do well in school.

Although his schoolwork was good, the social environment at school presented difficulties. Paul was rejected around age 7 to 8, and he thinks this was because of his appearance. He changed schools several times because of being unable to make friends. He thus developed a schema that people reject him, possibly because of his appearance. Years later, as an adult, he continued to feel that people would often reject him for being ugly, stupid, or weak. 

He stayed with his father but the relationship had multiple problems. The father was prone to extreme reactions and would say inappropriate things that Paul considered either stupid or embarrassing (or both). He became ashamed of his father, especially in public situations. Even alone at home, there were problems. Sometimes, the father would walk around the house naked, occasionally insisting on hugging the son, which made the son feel embarrassed and uncomfortable. 

Paul developed a schema from this troubled period of his family life that involved rejection and social disconnection, combined with shame and a broad sense that he has many faults. Withdrawing from others (social isolation) became a coping strategy, by which he made the best of bad situations. One positive aspect was that he came to be more self-sufficient, realizing he had to take care of himself because others could not be counted on to take care of him. 

As an adult, Paul has trouble with romantic relationships and feels them as full of pressure. Strong, assertive women are to be avoided, in case they become threatening like his father was. If a woman would get angry with him, it was a compelling signal for him to leave. Instead, he bonded with passive, dependent women, and the relationships took the form that Paul would make all the decisions. However, submitting to a domineering father did not prepare him to accept a strong manly role. He quickly ended most of these relationships because he was not able to be passive himself, as his schema dictated.

He had one lasting love relationship. It was with a depressed, passive woman who wanted him to take care of everything. She did not get angry with him, so he felt safe with her, and the relationship could start off well. However, he was never comfortable in the dominant role. Looking back, he said he had felt he had to take care of everything. He became angry with her, partly an overreaction to her dependency. However, he was not up to being the dominant one.

Recently, there had been another start to a relationship. This woman was more assertive than the other. She was willing and able to make decisions. Paul soon slipped into a highly passive role, consistent with his schema. Being passive and just going along was a defensive strategy that worked with his domineering father, so he easily adopted it with his assertive girlfriend, but she became bored with him (or so he thought, though it does seem likely). Her work sent her on foreign duty for six months. He says he decided to break it off when she went away, but she seemed to be fine with the decision to end it. The breakup triggered his rejection schema, and he confided some anger to his therapist. 

Now Paul says he has no energy and no desires. He is not good company even for himself alone. He is cold and arrogant and lacks spontaneity. He continues to do well at work, making no mistakes, trying to be smart and effective. However, he has difficulty responding properly when others praise and support him. He expects only criticism and rejection. He often feels ashamed and disgusted with himself. Often he compares himself to others and concludes “I am stupid.” He has developed social anxiety, which helps him keep his distance from other people. 

During therapy, he began to reach out to former relationship partners (both friends and lovers). However, they were not always receptive, particularly in cases in which they had genuinely liked him, but he rejected them based on his maladaptive schema. 

His biggest fear is being rejected, but when he is not rejected, he does not know how to act, so he often ends up rejecting the other. “I don’t know how to be with friends,” was a common comment by him in therapy. He said he did not know how to have a lasting good relationship. Peer groups in late childhood (ages 8–12) help to develop social interaction for many youngsters, but during that period, he did not develop those relationships and hence did not learn those social skills. 

When he was younger, he was good at playing guitar. However, he gave it up, partly because guitar playing brought him friendships and admiration from others, which was uncomfortable for him. His therapist encouraged him to resume playing guitar. He delayed doing this, making excuses such as that the guitars for sale in nearby stores were not just right for him. It was apparent that underlying anxiety about doing something to attract friends made him put off doing it. He made other excuses, such as that he had home repairs that needed to be done first. He said he did not know how to do the repairs, so he had to learn this first. He said he needs furniture but does not have a driver’s license or car, so he cannot get furniture because he cannot bring it home on the bus. All these were presented as his most pressing concerns and as supposedly valid reasons why he cannot busy himself with making friends and playing the guitar. 

**Concluding Interpretive Comment.** A schema of childhood rejection motivated Paul to cope with life by withdrawing from social intimacy while pushing himself to succeed at school. In adult life, he continued to be successful in work but continued to have bad relationships. 

This case involves a schema in which the self is defective, unwanted, inferior, or invalid in important respects. It evokes a sense that the person would be unlovable to others if they were to get to know him/her. Hence, rejection is an ongoing concern and indeed fear, even if others accept the person in the present. The person may become hypersensitive to criticism, rejection, and blame because all of these suggest future and even impending rejection. Side effects include self-consciousness (scrutinizing the self for flaws and faults that could lead to rejection). The person is chronically insecure, even in seemingly good relationships, because the schema says that sooner or later the partner will discover these faults and leave. One compares oneself to others frequently, as a way of pursuing adequacy and reducing the fear of being rejected. Sometimes, these comparisons reinforce the sense that oneself is inferior and flawed and hence undesirable as a relationship partner. There is a pervasive sense of shame about real or imagined faults in the self. The shame contributes to thinking that others will not want to sustain a relationship. 

The sense of self as deeply flawed makes it difficult to take action and make decisions. Anticipating that whatever one does will lead to disaster and hence rejection, the person may prefer to do nothing, thus remaining passive in order to avoid responsibility. There is a dubious but appealing sense that by doing nothing, one can postpone the moment at which one is abandoned by others because they will not see how bad the self is. However, in pragmatic terms, doing nothing is a vastly inferior strategy to some (not all) active, take-charge responses. 

Paul learned that he would be rejected by others and that is normal. That assumption can poison good relationships. If he finds himself in love or friendship with someone who accepts him and shows no sign of leaving, he finds this unfamiliar and uncomfortable and may break off the relationship.

### 4.3. Case Study 2: Julia Always Puts Others First (But Is Very Popular)

Julia, age 26, sought therapy for depression. Her first words to her therapist were “I feel overwhelmed”.

Her family background included an alcoholic mother prone to fits of screaming rage and a quiet (also alcoholic) father who had strong feelings himself but submitted to his wife. Julia also has a twin sister. The twin sister stood up to the domineering parents and resented Julia for submitting to them rather than supporting the sister’s resistance. Julia herself dealt with the difficult parents by focusing entirely on what they wanted, ignoring her own wants and needs, and endlessly striving for their approval. They showed little interest in Julia’s wishes and insisted that she follow their directions. As one example, Julia wanted to study anthropology and Japanese, but her mother said no and refused to pay for the costs of university study if Julia studied those things. She studied sociology, as her mother directed, but stopped in the third year.

In school, Julia made friends easily and was very popular. Her social success was helped by her pervasive self-abnegation. She always focused on the problems, desires, and concerns of other people. This kept them attached to her but at the cost of her not taking care of herself and her own life.

As one example, she mentioned that during a recent trip to Morocco, she had seen and bought some beautiful jewelry. The therapist remarked that she had never seen Julia wear any jewelry. She went about plain and unadorned, with her hair tied back in an unappealing fashion. Julia said she actually owned plenty of beautiful jewelry, which she loved. However, she never wore it because she was focused on other people and their problems, so she did not have time or energy to think about herself. At the next session, she showed up with her hair down and wearing some big, beautiful earrings. The therapist remarked with surprise that Julia was quite beautiful. However, in subsequent sessions, Julia reverted to her old pattern of wearing no jewelry and binding her hair. She was too busy to think about herself.

Her parents became sickly as they got old. Julia took it upon herself to care for them: “They need me,” she would say, consistent with her self-sacrificing schema. The therapist suggested that Julia’s younger brother and twin sister could share the burden of caring for the sick parents, but Julia dismissed that suggestion. She said her sister had bad feelings toward the mother based on their upbringing and so would probably not want to help. She said her brother was too young to be helpful. (In fact, he was 23 at the time). Again, she thought others need her, and the right thing for her to do was to sacrifice herself by providing care. She says she had been close to her father, but after she moved out of the parental home, maintaining a close relationship with him became complicated. Julia developed generalized anxiety. She tells the therapist that she has a strong feeling of guilt and that she has to make amends.

Julia never felt supported in what she loved. While young, she submitted to the verbal abuse of her mother (and her twin sister resented her for not opposing). She says it was necessary to sacrifice herself for the well-being of others. She suffers from not saying what she wants. She affirms “I have the right to have my life. You can’t insult people and expect people to continue to be nice”. Furthermore, she continues to devote herself to others. She feels guilty for going wrong and for hurting people. She surrounds herself with people who are not doing well and feels guilty for not being able to help them more. She thinks people resent her for not being totally available because she feels a deep obligation to do whatever people expect of her. She herself has had interests and passions that she was not able to realize (such as when her mother dictated her course of study). 

Romantic relationships are difficult. Getting love for her must be a test, in which she does everything for the man. Hence, she is mainly attracted to men who initially are not attracted to her and who have problems and want to be helped. She strives to prove that she can please them and help them. She describes her early love relationships as “toxic”. As she recalls, these men paid little attention to her wishes or her happiness. She says they abused her generosity: “I gave them so much, but they never gave me as much”. The therapist asked whether Julia’s purpose in giving was to elicit reciprocal giving from them. Julia quickly said no, but the therapist found the denial less than fully convincing, given how Julia repeatedly expressed disappointment over the lack of reciprocation. 

Eventually, she did find a relationship with a man who loved her and treated her well. At one point, his work required him to move to Italy for six months. With the impending separation, she told him that love for her meant that she would give herself completely to him, that she would give him everything she is, and that all of her life and love would be for him. The man started to cry and said he was not prepared to want or accept such total self-abnegating love. As a result, they broke up. She was very sad. The therapist asked why she had put such an extreme condition on continuing the relationship. Julia said it was better this way because now she would have some time for herself.

When she was asked whether she had expected the young man to accept her love under those extreme conditions, Julia admitted that she had not. She said she had learned that it is a problem for other people that she gives herself so totally. Thus, she had some insight into how her way of relating to others is counterproductive. However, whenever she enters a relationship, she reverts to this schema of denying herself and focusing entirely on the other. 

**Concluding Interpretive Comment.** A self-abnegation schema caused Julia to always put others ahead of herself. This schema helped to make her popular, but in adulthood, she misses opportunities, has difficulty making decisions for herself, and allows others to take advantage of her. 

As with Paul, this case involves a schema in which the self is defective; however, Julia’s adaptation is self-sacrifice rather than Paul’s isolation. This corresponds to an excessive importance given to the needs, desires, and reactions of others, at the expense of one’s own needs in order to obtain their affection or their approval, for fear of being abandoned or to avoid reprisals. The family origin of this schema must be sought on the side of an affection that comes under the conditional: to be loved by the parents, to obtain their approval, the child represses his or her natural tendencies. The exaggerated concern to always consider others before oneself is voluntary. The reasons are usually linked to the fear of hurting others, to avoid feeling guilty of selfishness, or to maintain contact perceived as necessary to others. Julia did sometimes complain that her own needs were never met, resulting in resentment towards others.

Julia experiences frequent depressive episodes because she has no control over the events of her life. Her only way to experience control is to engage in toxic (romantic) relationships, re-enacting aspects of the toxic relationship she had with her mother. The self-sacrifice was adaptive in order to avoid conflicts with her mother (conflicts were frequent between her mother and her twin sister). By responding favorably to her mother’s needs, she could obtain a few moments of kindness from her. However, in adulthood, this tendency to sacrifice herself for others leads her to develop (romantic) relationships for which her self-sacrifice will have no limit. This behavior drives away the men she seeks to form a relationship with. 

### 4.4. Case Study 3: Carol Learns to Trust Nobody (But Develops Autonomy)

Carol came for therapy for anxiety. She was 23. In the first session, she expresses rumination and jealousy as well as anxiety. She quickly explains that her mother died of cancer when she was eight. She did not know her father, so upon her mother’s death, she went to live with her maternal aunt. Then, her father came back into her life and took her into his home, with his new family. Although he accepted his parental responsibilities, he was not very interested in Carol and did not take good care of her. She had little in common with her father’s new family, and relations deteriorated quickly. She slept in a bedroom in the basement and ate her meals alone. She felt neglected and formed a schema that people neglect her. She coped with that by striving to be self-sufficient, independent, and autonomous. As an adult, she still has these patterns. She is good at being independent, autonomous, and self-sufficient, so she does not need intimacy with others.

She stayed with her father’s family for five years, roughly from age 9 to 14. She was trying to get noticed. She felt nonexistent. She had an anxiety and depression syndrome with several suicide attempts (especially in times of change). She had been a very good student, but during this period, she became a very poor one and briefly dropped out of school. She thought that doing badly at school might get others concerned about her and interested in her. She also developed anorexia, which seemed linked to a desire to disappear. None of this succeeded in getting others to care about her. This was when she started to focus on becoming independent. She decided that rather than trying to attract others to be interested in her, she would be self-sufficient and not need others. At this point, she stopped being anorexic and she went back to school, even resuming her previous good performance. 

After five years she moved back in with her maternal aunt, but the relationship remained difficult. Her time with her father and his family led to a pervasive fear of being neglected. In her aunt’s home, she felt the same difficulties with the children of her aunt. She says she helped the aunt a great deal with the housework and chores, while the aunt’s children never helped. They said to her, “it’s not your home”, “you are not our sister”. Later, she learned that the aunt had gone to court to insist on Carol’s returning to her because the father had neglected her seriously. However, history repeated itself with the aunt’s children. She felt betrayed.

In therapy, Carol spoke very little, and when she did speak, her voice was so quiet as to be nearly inaudible. Surprisingly, however, Carol disclosed that she sometimes entered speaking contests and performed rather well, giving clear and entertaining presentations. She liked the contests because they allowed her to be listened to, to make people laugh, and in general to attract attention. She never won but sometimes came in second or third. She did well enough that her school asked her to be a student coach for speaking contests. She accepted and was proud to be coaching other students. She liked having people listen to her.

A pattern dating from this time and continuing in the present is a reluctance to accept compliments. When people tell her she is doing well in her studies or is attractive or whatever, she becomes uncomfortable and wonders why they are saying this. She thinks it must be manipulative. It cannot be that others think she is good and simply tell her so.

Thus, one key theme is that she felt mistreated when living with her father during her preadolescent years. She formed a schema that other people would not treat her well, would betray her, and that others cannot be trusted. When forming a new relationship with a seemingly good man, she applies this schema, expecting the man to betray and abuse her. Pervasive mistrust makes an intimate relationship difficult.

Two years earlier, at age 21, Carol started living with a man. She felt she could not trust him. She was afraid that he was hiding things from her and that he will not tell her everything. She is very jealous and suspicious. She sneakily looks at his phone to see whether he communicates with other women. She tries to resist the urge to check his phone, but she says the urge is too powerful, and she cannot resist. When he sees her doing that, he asks her why, and she answers that she sees he has communicated with another woman (who actually is only a friend, not a lover). To her, that proves she is right to distrust him. 

He would go to take a shower and leave his phone on the table. The therapist points out that he presumably had nothing to hide. If he were having illicit or clandestine phone communications with other lovers, he would not leave his phone there, especially given that he had already caught her looking at his phone previously. Nevertheless, she continued to look, and when she saw any communication with any other woman, she felt that her distrust was validated. As far as can be ascertained, he was faithful to her for the years they lived together, but she was certain he must have had someone else because she thought no one could be sufficiently interested in and satisfied with her. 

Carol explains that she learned from mutual friends that when he was a university student, he dated multiple women. This seems to have been a fairly normal part of looking for a partner, but Carol regards it as a reason to distrust him as potentially unfaithful. The therapist asks her why she chose to live with a man she cannot trust, a man whom she suspects of having others. However, this is the schema for relationships from when she grew up, with her father more interested in his new family than in her. Very likely, her schema for what relationships are like, based on her preadolescent time living with her father and his new family, was a reason that she became involved with this man despite her not trusting him.

Her distrust cramps her social life. She would refuse to go to a party without him, and likewise, she declines opportunities to visit family or relatives, because she does not want to leave her boyfriend alone. She thinks he will see other women if she is gone for any length of time. All this is quite stressful for her. After several years, she decided to break up with him (while she was in therapy). 

At one point in therapy, she says that she can never trust anyone. This was a strong belief. The therapist asked Carol whether she believed in her (the therapist). Carol replied, “Yes—now”. 

Carol decided to become a lawyer. This would be a way to help people. Her attitude was that she cannot trust others to be good to her, but she still wants to connect with others. For that, she desires power and influence. She liked the public speaking contests because they offered a way to connect with other people (the audience), to influence them. Her ambition to be a lawyer fits this also: As a lawyer, she can become involved with other people and can use her skills and powers (such as for effective public speaking) to help them, but not require that they care for her. She would not have to trust them to be concerned about her feelings, because she works for them.

Carol always takes a long time in the morning to get dressed. She changes clothes several times before going out, hoping the right clothes will help her to be noticed at school. However, this ritual generates a lot of stress.

**Concluding Interpretive Comment.** Disrespect and betrayal by her parents created a pattern of mistrust that led Carol to develop her autonomy and independence (she cannot rely on anyone). Once an adult, this schema makes the woman reluctant to reveal herself to others, thus handicapping adult relationships. 

To avoid finding herself in situations where she might feel humiliated or neglected, she adapted by learning to rely only on herself to the point of obsession (e.g., the obsessive care taken with her dressing). Her early attempts to get others’ attention included failing in school or developing eating disorders such as anorexia. However, faced with the ineffectiveness of these strategies, she opted for autonomy. The lack of reassuring interpersonal relationships turned into a generalized anxiety disorder in adulthood when she had to develop intimate relationships. The other is seen as a potential threat that she must guard against. She expects others to hurt her, abuse her, humiliate her, lie to her, or take advantage of her. It usually involves the perception that the harm is intentional or the result of wrongful negligence. 

The feeling that the betrayal of others will always prevail pushes her to reveal nothing about herself. The less others know about her, the less inclined they will be to use this information against her but to the detriment of harmonious relationships. Being silent avoids communication and fosters misunderstandings. However, what is the point of discussing when you are convinced that the other will lie? The choice to not communicate leads to considering the person suspected of lying as responsible for her unhappiness and preventing the relationship from developing. 

She learned that others’ neglect of her was normal. The mistrust she has developed toward these people (and, by extension, to nearly everyone) makes her interpersonal (love) relationships impossible.

## 5. Discussion

The impact of meaning on mental health is often mediated by schemas. We have provided three case studies in which an individual formed a general schema during preadolescence as a way of coping with a difficult, stressful family environment, which led to mental health problems and interpersonal failures during adulthood. 

We assume relationship schemas can form at any age, but the most influential ones may form during the preadolescent years (8–12, roughly). The three case studies all struggled with atypical, difficult, unsatisfying, and sometimes cruel social environments during that age period. They developed meaningful interpretations of their social environment and its relationships, which seemed to them to be the way the world and social life in general will be. 

These schemas are an important part of life’s meaning for these individuals. While a given person’s life usually draws meaning from multiple sources [13], relationships are generally among the most important. These older children, approaching the passage of adolescence when suddenly their desires and impulses will change and increase, are learning how to have a partner relationship with someone, as opposed to just being the child and letting the parents run things. Learning all this from a bad social environment creates a schema that is not suited for a long, happy marriage. 

Paul learned a schema that assertive relationship partners are dangerous and that coping with passive withdrawal, along with high performance in school and work, was the optimal strategy. In adulthood, he sought out only passive, non-threatening partners but found himself wishing to be passive. Julia’s schema involved coping with demanding partners by focusing entirely on what she could do for them, denying her own wishes, and submitting to their demands. In adulthood, she sought out needy partners but found that some partners were uncomfortable with her sweeping self-abnegation and self-sacrifice, and she was also unhappy with their failure to reciprocate. Carol learned that others neglect and abuse her, so her schema involved pervasive distrust of others and cultivating self-sufficiency. In adulthood, she could not accept compliments, undermined her romantic connection by pervasive, unfounded suspicions, and suffered frequent anxiety over how to make herself attractive to others. 

To be sure, our work has clear methodological limitations, and we look forward to future work that may augment, confirm, and/or correct our findings. The present article is based on a very small sample drawn from intensive, ongoing psychotherapy, and its methods involve qualitative interpretation of three personal narratives. The three were chosen as fairly typical problems seen in the first author’s clinical practice, but generalizing should be done with extreme caution, especially to individuals in other, different cultures. Our work should be regarded as an exploratory early step in seeking to understand how specific meanings can have an impact on mental health. 

### 5.1. Implications for Theory

One of the broad questions about the psychology of meaning is what form it takes in individual lives. People can often give detailed, coherent narratives about their lives; when asked to articulate a general principle, they typically can give only vague answers, the most common of which is family [14,15,16]. All three of the case studies we covered were heavily influenced by family relations, but the resulting meanings in their lives were hardly ideal. Nor were they a strong foundation for how to behave in adult relationships. The damage in terms of mental health was unmistakable in these cases. Thus, dysfunctional or maladaptive schemas are possibly a major avenue by which meaning has an impact on mental health.

A schema, as we have used the concept, is a kind of generalized narrative. It is acquired from an actual series of events, such as occurring in family interactions. Narrative sequences of events occurring during preadolescence may be particularly potent in shaping how the person comes to understand interpersonal relations generally. The stories experienced during that period create a set of assumptions and expectations that are later applied to adult relationships. 

A broad issue is what makes a schema dysfunctional. We emphasize that the schema was formed in each of these cases to deal with family relationships that were difficult and hurtful. Applying the same schema to subsequent adult relationship partners can lead to problems in multiple ways. Most obviously, even a perfectly wonderful adult relationship partner would have difficulty forming a relationship with someone like these patients who insist on interpreting their actions as recapitulating the sort of hurtful treatment they received previously. Moreover, the very selection of adult relationship partners may be distorted by past experiences, such that people make suboptimal choices (thus further reducing the chances of finding the perfectly wonderful one) of partners who seem familiar precisely because they recapitulate the sort of troubled close relationship that the person had as a preadolescent. A related possibility is that while learning to cope with a difficult relationship as a preadolescent, one also failed to learn how to conduct a healthy, mutually satisfying relationship—a failure and lack that again would be detrimental to interpersonal success in adulthood. 

Thus, one may learn in preadolescence that others who supposedly love one will reject or betray one. To cope, one learns to prevent rejection by being alone (Paul)—or by being totally giving of oneself (Julia). One learns to prevent betrayal by being pervasively suspicious of others (Carol). Each of these schemas damaged the patient’s adult attempts at intimacy.

While the dysfunctional and maladaptive aspects of the schema are apparent in adulthood, it is useful to keep in mind that the schema was originally a way of coping with a difficult situation, and in that respect, it has, or at least had, positive and adaptive aspects. In addition, even though the problematic aspects of the schema damaged adult relationships, there were some lasting benefits. Paul became highly successful at his work. Julia became popular. Carol became self-sufficient and developed public speaking skills that may help her excel as a lawyer.

### 5.2. Implications for Therapy

Improving mental health in therapy involves multiple steps. The patients must first become aware of what their dominant schemas are and how they are using them to shape their adult relationships—as well as why the application of these schemas ends up undermining relationships that could otherwise be successful and satisfying. They must not only understand the schema but also how it is instantiated in their everyday interactions. 

Clinical observations suggest that patients are often dismayed to realize how their schema has been altering their behavior in ways that damage important relationships. It is useful to point out that the schema was indeed an adaptive way for them to cope with their earlier, difficult situation, and so it was originally quite successful. They can feel reassured or even proud of this adaptation. The challenge is to learn alternative, better ways of forming adult relationships. Another useful point was to point out the positive outcomes of the schema along with the negative ones.

Therapeutic progress with these part patients proceeded roughly along these lines. Once the patient understood the schema, the therapist instructed the patient to make a list of behaviors that enacted or sustained that schema in the current relationship. These lists were fairly easy to produce (and often rather long). Next, the therapist encouraged the patient to change some of these specific behaviors and see what happened as a result. Sometimes the patient initially expressed confusion about how to change, especially because the typical responses were automatic. However, when asked what alternative action would work, the patient could usually articulate some good solutions, which the therapist said to try out in the near future (e.g., before the next session). Often, the patient was surprised to find that nothing much changed as a result of the alteration in the patient’s behavior. The relationship continued and sometimes improved. For example, Julia discovered that her friends remained her friends even when she stopped trying to solve all their problems at the expense of her own well-being.

## 6. Conclusions

In previous research, much has been learned by measuring subjective ratings of the quantity of meaning in life, and we encourage researchers to continue using that approach. In this article, in contrast, we have undertaken a different approach that we think has the potential to expand and inform the study of meaning in life. It considers specific meaning structures (schemas) by which people come to understand their social world. People may differ not only in the quantity of meaning but also in the types of specific meanings by which they make sense of self and the world.

## Data Availability

No data are publicly available.

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
