# Peer review of "Dysfunctional Schemas from Preadolescence as One Major Avenue by Which Meaning Has Impact on Mental Health"

_ijerph, 2023, doi:10.3390/ijerph20136225_

Round 1

Reviewer 1 Report

Submitted manuscript well illustrated the dysfunction of meaning or meaning of life with focusing on three cases. If only a few things are supplemented, it can be evaluated as a manuscript that can be published in journal. What I think needs to be improved is:

 1. The subheadings should be a bit more specific.

 2. Of course, this manuscript is not structured like a traditional case study, but case are presented. If so, at least a summary of the methodological aspects of data collection and analysis should have been presented. Even if they were the first author’s clinical cases, not the cases planned for research. If so, those things should be described as well.

 3. It is also related to what I described above, but it should have been stated that these cases were informed that their stories would be published in article and that permission was obtained.

4. Reference writing style is not suitable for this journal.

This manuscript was well summarized and described for readability.

Author Response

Response to Reviewer 1 Comments

Submitted manuscript well illustrated the dysfunction of meaning or meaning of life with focusing on three cases. If only a few things are supplemented, it can be evaluated as a manuscript that can be published in journal. What I think needs to be improved is:

  1. The subheadings should be a bit more specific.

              Thank you. We made several subheadings more specific.

  1. Of course, this manuscript is not structured like a traditional case study, but case are presented. If so, at least a summary of the methodological aspects of data collection and analysis should have been presented. Even if they were the first author’s clinical cases, not the cases planned for research. If so, those things should be described as well.

              We have added a paragraph to describe the methodological and therapeutic process. This is at the beginning of the section on case studies, just before the first case.

  1. It is also related to what I described above, but it should have been stated that these cases were informed that their stories would be published in article and that permission was obtained.

              Good point, we have added the explicit statement that the patients signed up for therapy and furnished written consent that information about their case could be published anonymously in a scientific or clinical journal.

  1. Reference writing style is not suitable for this journal.

              We have checked the reference formatting rules and fixed the reference reporting style. We apologize for the small deviations from the usual style.

This manuscript was well summarized and described for readability.

              Thank you!

Reviewer 2 Report

The topic proposed in the article is interesting and it could represent a thought-provoking contribution for the professionals of the clinical field, but as it is now, it presents some critical points.

The title contains a statement according to which “dysfunctional schemas are the major avenue by which meaning has impact on mental health”. Such a statement sounds as a generalization and as such it should be proved with a robust methodology, which is not the case of the presented article.  The authors’ interpretation of the behaviors and distress of the three clients described in the article, is certainly very interesting and somehow convincing, but such interpretation is just one possible interpretation among the many that could be given from other points of view. Moreover, the cases presented are only some examples of schemas that one can develop in his/her life; many others could be developed according to the personal experience. This should be underlined. Unless the authors chose those specific cases for some specific reason that should be expressed? 

So, I suggest changing the title in “The impact of dysfunctional schemas on mental health”. And make clear that the cases presented serve as examples of how it could work the theory of Schemas to explain persons’ distress or dysfunctional behaviors, not as “the truth”.

Furthermore, the article doesn’t add nothing to the theory, in fact, the paragraph “Implication for theory” doesn’t add anything to what already claimed in the introduction.  In this sense, it appears tautological, the authors start from a piece of theory, describe how it could work in practice and then they conclude that the cases illustrated confirm the theory described since the beginning. Yet the purposes of the article expressed in the introduction seemed to have a broader development.

In the introduction, the authors claim: 

 We focus particularly on schemas that can be adaptive ways of coping with an atypical and stressful interpersonal environment — but which prove maladaptive when subsequently applied to adult relationships. A clinician can help guide interpretation and restructuring of the schema, possibly even transforming or replacing it, so as to foster behavior change and improve mental health (Italic is mine).

Besides the awkward wording of the sentences (the English language should be revised throughout the whole article), of the two points there indicated, only one has been addressed (the focus on schemas that can be adaptive ways of coping with an atypical and stressful interpersonal environment — but which prove maladaptive when subsequently applied to adult relationships.)  In the paragraph “implications for therapy”, the authors make a general claim on how the theory of Schemas could be helpful in therapy without saying howtherapists could guide clients to reinterpret and restructure the clients’ schemas.  I think that a case study article would gain in depth addressing the therapeutic issue. I suggest the authors to add in each case what the therapist did, what method of restructuring was used to help the client to overcome the dysfunctional schema. Without this the article sounds incomplete.

English language should be revised throughout the whole article

Author Response

The topic proposed in the article is interesting and it could represent a thought-provoking contribution for the professionals of the clinical field, but as it is now, it presents some critical points.

Thank you! We hoped to make a thought-provoking contribution. We have sought to emphasize the critical points.

The title contains a statement according to which “dysfunctional schemas are the major avenue by which meaning has impact on mental health”.

Thank you for noticing this. We did not intend to say schemas were the major avenue, nor to exclude or disrespect other ways in which meaning can affect mental health. We have revised the title to be clearer that we think dysfunctional schemas are one of the major avenues.

We also note that the Discussion articulates our position in perhaps a less ambiguous fashion: “dysfunctional or maladaptive schemas are possibly a major avenue by which meaning has impact on mental health.” We hope this dispels any lingering concerns that we claim too much.

Such a statement sounds as a generalization and as such it should be proved with a robust methodology, which is not the case of the presented article.

We agree. Again, we did not mean to disparage other ways in which meaning can influence mental health.

  The authors’ interpretation of the behaviors and distress of the three clients described in the article, is certainly very interesting and somehow convincing, but such interpretation is just one possible interpretation among the many that could be given from other points of view. Moreover, the cases presented are only some examples of schemas that one can develop in his/her life; many others could be developed according to the personal experience. This should be underlined. Unless the authors chose those specific cases for some specific reason that should be expressed?

Again, we regret having given the impression that we think these illustrate the only way that meanings and schemas can affect mental health. We agree that there are multiple schemas that people can develop and, moreover, that there may be other dysfunctional schemas that could influence mental health in other ways. In fact, it is our hope that by opening up this approach, we would encourage others to come forward and supply some of these other schemas and influences.

So, I suggest changing the title in “The impact of dysfunctional schemas on mental health”. And make clear that the cases presented serve as examples of how it could work the theory of Schemas to explain persons’ distress or dysfunctional behaviors, not as “the truth”.

We appreciate the suggestion. However, please note that the suggested title is even more comprehensive than the one we used and hence could elicit the same criticism of claiming too much. So, instead, we toned down the title to say that dysfunctional schemas are (only) one of the major ways by which meaning can have an impact on mental health.

Furthermore, the article doesn’t add nothing to the theory, in fact, the paragraph “Implication for theory” doesn’t add anything to what already claimed in the introduction.  In this sense, it appears tautological, the authors start from a piece of theory, describe how it could work in practice and then they conclude that the cases illustrated confirm the theory described since the beginning. Yet the purposes of the article expressed in the introduction seemed to have a broader development.

Thank you for this comment. Nevertheless, there are two important points brought up in the Discussion that are not in the introduction. One is that children or adolescents usually cope effectively with bad events. The second is that even in difficult situations and problem relationships, people develop other skills that can be useful both in therapy and in adapting to life as an adult.

In the introduction, the authors claim:

We focus particularly on schemas that can be adaptive ways of coping with an atypical and stressful interpersonal environment — but which prove maladaptive when subsequently applied to adult relationships. A clinician can help guide interpretation and restructuring of the schema, possibly even transforming or replacing it, so as to foster behavior change and improve mental health (Italic is mine).

Besides the awkward wording of the sentences (the English language should be revised throughout the whole article), of the two points there indicated, only one has been addressed (the focus on schemas that can be adaptive ways of coping with an atypical and stressful interpersonal environment — but which prove maladaptive when subsequently applied to adult relationships.)  In the paragraph “implications for therapy”, the authors make a general claim on how the theory of Schemas could be helpful in therapy without saying howtherapists could guide clients to reinterpret and restructure the clients’ schemas.  I think that a case study article would gain in depth addressing the therapeutic issue. I suggest the authors to add in each case what the therapist did, what method of restructuring was used to help the client to overcome the dysfunctional schema. Without this the article sounds incomplete.

We agree that in the long run it would be helpful to therapists to know how best to treat particular problems, including those based on dysfunctional schemas. Given the space limitations as well as our expertise, the goal of this article has been only to illuminate how dysfunctional schemas can form and lead to interpersonal problems and mental illness. The second author is not a therapist at all and the first does not feel ready to dictate how other practitioners should practice. The goal of this article is thus focused on explaining how meanings can contribute to mental illness (which is the focus of the special issue), not to elucidate therapeutic techniques. That could well be a valuable contribution but would take considerably more evidence and of a different kind. Understanding is probably a vital first step. Nevertheless, we have added a long paragraph to the “Implications for Therapy” subsection in the Discussion that describes how the therapist used the schema understanding to beneficial effect in these three cases. We hope that is a suitable first step toward illuminating one way of making use of the enhanced schema understanding in therapy.

English language should be revised throughout the whole article

              We do not fully understand what or where the problem is. The first author is indeed a French citizen for whom English is not the native language, but she has published many articles in English. The second author is a native English speaker, indeed with over 700 publications in that language. While we agree that writing can nearly always be improved, we would appreciate more specific articulation of what improvements might be sought. Because the reviewer says the problem affects the entire article, let us start with the first paragraph. What is unclear?

The issue of meaning in life was once considered too abstract, subjective, and unscientific for scientists to study (most obviously in the days of rat labs). In recent decades, researchers have developed tools and theories to study it. The vast majority of such research has involved measuring (or occasionally manipulating) a person’s global feeling of meaningfulness in life. Essentially, it compares people who report their lives as highly meaningful against those who don’t, typically with one number on a scale to represent the person’s perceived meaningfulness for his or her entire life. Meaning is treated as a quantity, so roughly all meaning is the same in that sense. All lives have the same kind of meaning, when it is measured as a single number, or even a few. While we fully respect and appreciate the contributions of that approach, this paper takes a different and more nuanced approach. We examine how specific meanings form as ways of coping with difficult relationships or other life circumstances — and how the resulting schemas can hamper adult functioning.

Reviewer 3 Report

This article focused on the schema as an important form of meaning, and examined how the dysfunctional schemas formed and hamper adult mental health. Compared with the traditional method by using scale to represent one’s perceived meaningfulness for his or her entire life, this article introduced a different and more nuanced approach, i.e., case study, which may help us to understand the impact of meaning on mental health in adults. 

I have some comments for consideration.

1. Though these three cases are independent, there are some similarities. For example, the family, the response style of pressure in preadolescent, and the impact of dysfunctional schemas. The authors should conclude the nature of dysfunctional schemas in the discussion part.

2. The title is a little general, it should be more specific. The title can be changed into “Dysfunctional schemas in preadolescent as major avenue by which meaning has impact on mental health”.

3. Change the key word “childhood” into “preadolescent”.

4. According to the ethical norms in clinical psychotherapy, the authors should mention whether the names of three case participants was anonymous or replaced for confidentiality? And whether the three participants were acknowledged that their life experiences were shared in this article. This is very important!

5. The Theory part seemed to introduce how the schemas forme and influence human's mental health, instead of the theory. The title "Theory" should be changed to cover the content.

 The Quality of English Language is good.

Author Response

This article focused on the schema as an important form of meaning, and examined how the dysfunctional schemas formed and hamper adult mental health. Compared with the traditional method by using scale to represent one’s perceived meaningfulness for his or her entire life, this article introduced a different and more nuanced approach, i.e., case study, which may help us to understand the impact of meaning on mental health in adults.

I have some comments for consideration.

  1. Though these three cases are independent, there are some similarities. For example, the family, the response style of pressure in preadolescent, and the impact of dysfunctional schemas. The authors should conclude the nature of dysfunctional schemas in the discussion part.

We gave considerable thought to this and decided it is a profound and important suggestion. We have added a paragraph in the “Implications for Theory” subsection of the Discussion to tackle head-on the question of why a dysfunctional schema is dysfunctional. Thank you.

  1. The title is a little general, it should be more specific. The title can be changed into “Dysfunctional schemas in preadolescent as major avenue by which meaning has impact on mental health”.

Thank you for the suggested title! We have adopted almost exactly that. (We added the word “one” to foreclose the kind of misunderstanding Reviewer 2 noted.)

  1. Change the key word “childhood” into “preadolescent”.

Good idea! We did.

  1. According to the ethical norms in clinical psychotherapy, the authors should mention whether the names of three case participants was anonymous or replaced for confidentiality? And whether the three participants were acknowledged that their life experiences were shared in this article. This is very important!

Yes, correct. We have made this change.

  1. The Theory part seemed to introduce how the schemas forme and influence human's mental health, instead of the theory. The title "Theory" should be changed to cover the content.

Yes, we replaced by “Acquisition of dysfunctional schemas”.

Reviewer 4 Report

In my opinion, the paper “Dysfunctional schemas as major avenue by which meaning has impact on mental health” shows really too many shortcomings in many different aspects to be published. 

The introduction does not provide sufficient background and does  not include the most relevant references. The references provided are really very scarce. There are only vague references to meaning, schemas, etc. There is no research but only the presentation of three clinical cases. The cases are only described without any theoretical or methodological frame of references concerning how they were treated ( for instance which kind of therapeutic approach? How long was the therapy?)

Scientific soundness, quality of the presentation, and significance of content are really almost absent.

Author Response

In my opinion, the paper “Dysfunctional schemas as major avenue by which meaning has impact on mental health” shows really too many shortcomings in many different aspects to be published.

The introduction does not provide sufficient background and does not include the most relevant references. The references provided are really very scarce. There are only vague references to meaning, schemas, etc. There is no research but only the presentation of three clinical cases. The cases are only described without any theoretical or methodological frame of references concerning how they were treated (for instance which kind of therapeutic approach? How long was the therapy?)

As noted above in response to Reviewer 1, we added a section called “Method” just prior to the presentation of the case studies. It explains how the data were obtained and what the therapeutic approach was.

Scientific soundness, quality of the presentation, and significance of content are really almost absent.

It is difficult to make specific changes in response to this review. We wish the reviewer would have indicated what the most relevant references (especially ones we omitted) would be. The paper is based on case studies, and we contacted the special issue’s editors before writing the paper to ensure that case study presentations would be welcome as an empirical basis for a contribution to the special issue.

Round 2

Reviewer 2 Report

NO COMMENT

Author Response

The reviewer reported "No comment".

Reviewer 4 Report

Some changes were made to the paper. However, I think that some more work need to be done before publishing.

First the authors must underline that they  use "qualitative vignettes" to support their hypotheses. Maybe a paragraph devoted to how this kind of work helps to add information. Also I think a paragraph about the study's limitation needs to be added.

Finally two more specific details? Preadolescence needs to be defined in a more specific way? Why schema develops in this period. Second in the conclusions the authors talks about measuring. They are not measuring they describe and interpret. 

Author Response

Here are the responses to the reviewer 4.

1. Some changes were made to the paper. However, I think that some more work need to be done before publishing. First the authors must underline that they use "qualitative vignettes" to support their hypotheses. Maybe a paragraph devoted to how this kind of work helps to add information. Also I think a paragraph about the study's limitation needs to be added.

We respectfully submit that the term “qualitative vignettes” is not an accurate description of what we did. Research that is presented as using vignettes typically uses fictional, imaginary, or hypothetical stories and collects data from participants who read them and provide ratings. Vignettes can be systematically altered by researchers to compare how people rate different versions with specific, well-controlled variations. While we respect that method, it is not at all what we did. Our cases are actual case histories from ongoing therapy.  

We have however added the point that our work in this case is purely qualitative (see the new fifth paragraph of the Discussion). That paragraph also makes explicit the limitations of the study, as the reviewer requests, and we agree that noting such limitations is very appropriate.

2. Finally two more specific details? Preadolescence needs to be defined in a more specific way? Why schema develops in this period.

We defined preadolescence and added some comments concerning why schema could develop during this period. We appreciate the suggestion and have sought to respond to it, but we do note that the preadolescence issue is not central to our theory or analysis. The purpose of our paper is to advance understanding of how dysfunctional schemas are an avenue by which meaning contributes to mental health problems. For that, it would be fine if the schemas originated in some other phase of life, and in some cases they probably do. We merely have observed that in these and most other cases we have seen, the schema seems to have its origin during preadolescence. A plausible reason is that this stage of development has been regarded as an important one for learning how to conduct interpersonal relationships.

3. Second in the conclusions the authors talks about measuring. They are not measuring they describe and interpret.

Our phrasing here seems unfortunately to have misled the reviewer, and it would likely mislead others as well, so we thank you for calling our attention to it. We intended the opening sentence to convey respect for prior work that has relied in simple ratings to measure someone’s global self-appraisal of meaningfulness. This was to provide a contrast with the approach we have taken. The point of the conclusion is that we do not criticize prior work or seek to replace those methods. Rather, we seek to provide an alternative approach to augment what can be learned. We have added a few words to the Conclusion to make this clearer and more explicit.
